# Learning and memory in the orange head cockroach (*Eublaberus posticus*)

**Christopher A. Varnon** *, Erandy I. Barrera, Isobel N. Wilkes

Department of Psychology, Laboratory of Comparative Psychology and Behavioral Ecology, Converse University, Spartanburg, South Carolina, United States of America

* Christopher.Varnon@Converse.edu

## Abstract

This paper describes two experiments aimed at establishing the orange head cockroach (*Eublaberus posticus*) as a model organism for behavioral research. While many invertebrate models are available, cockroaches have several benefits over others that show impressive behavioral abilities. Most notably, cockroaches are long-lived generalists that can be maintained in controlled indoor laboratory conditions. While the most popular cockroaches in behavioral research, *Periplaneta americana* and *Blattella germanica*, have the potential to become domestic pests, our *E. posticus* is extremely unlikely to escape or infest a human environment, making it a very practical species. In our first experiment, we investigated the ability of *E. posticus* to associate novel odors with appetitive and aversive solutions. They quickly learned to approach odors associated with a dog food sucrose solution and learned to avoid odors associated with salt water. The second experiment repeated the methods of the first experiment, while also testing retained preferences for conditioned odors, from 15 to 1,215 minutes after the conditioning procedure ended. We found that preferences for odors associated with food were strongest 45 minutes after training, then decreased as a function of time. Our work is the first to show associative learning and memory in the orange head cockroach. Findings are discussed in comparison to other invertebrate models as well as to other cockroach research.

**Data Availability Statement:** All relevant data are within the paper and its Supporting Information files.

## Introduction

The purpose of this paper is to establish the orange head cockroach (*Eublaberus posticus*) as an insect model of associative learning. Insects are useful model organisms to study the ontogeny and phylogeny of behavioral processes [1–4]. Two prominent models are the fruit fly, *Drosophila melanogaster*, and the honey bee, *Apis mellifera*. While work with *Drosophila* has a major focus on genetics [5, 6], they are also an important model for learning and memory [7, 8]. Honey bees have received recent attention as neurophysiological models [9–11] but are also a very popular model of complex learned behavior and are the most popular insect researched in comparative psychology [12]. Psychological topics on honey bees are diverse, ranging from learned helplessness [13] to abstract concept learning [14].

**Funding:** This project was supported by grant P20GM103499 (SC INBRE) from the National Institute of General Medical Sciences, National Institutes of Health. The funders had no role in study design, data collection and analysis, decision to publish, or preparation of the manuscript.

**Competing interests:** The authors have declared that no competing interests exist.

Surprisingly, comparatively little attention has been given to cockroaches, despite several benefits that would make them excellent models. First, cockroaches have a relatively long lifespan, with many species living over a year [15, 16]. *Drosophila* only live 8–14 weeks [17]. Queen honey bees can live 1–2 years, but worker bees, the primary subject of behavioral research, live only 2–6 weeks [18]. Second, while *Drosophila* and bees have a distinct larval stage, cockroaches do not undergo complete metamorphosis [19, 20]. Instead, the juvenile cockroach resembles an adult, allowing for similar psychological procedures to be conducted throughout the lifespan. Third, cockroaches are social generalists, while bees are eusocial insects with a highly specialized ecology. The specialized colonial nature of bees, which involves not only distinct castes, but distinct roles within castes [21], may be a contributor to some of their impressive behaviors, including cooperative foraging [22, 23]. However, this degree of specialization may also lead to challenges in other areas, like acquiring conditioned taste aversions [24]. Alternatively, the social generalist nature of cockroaches [25] makes them similar to a rodent model. While they have a range of social behaviors [26, 27], each individual is well-equipped for its' own survival. Investigations with both bees and cockroaches may lead to interesting comparisons between specialists and generalists, or between eusocial and social life strategies. Finally, most cockroaches can easily be maintained year-round in controlled indoor facilities. This is standard for *Drosophila* work, but is not possible for honey bees, leading some laboratories to adopt bumble bees (genus *Bombus*) as indoor alternatives. Unfortunately, artificial conditions may even prevent bumble bees from thriving and being practical subjects [28, 29]. Taken together, we believe that cockroaches may be an ideal model to complement existing insect models of behavior, and development of a cockroach model would also support much needed model diversity [12, 30].

We are not alone in our interest in cockroaches as behavioral models. A growing body of research demonstrates they have excellent learning abilities. Topics include classical conditioning [31–33] operant conditioning [34, 35], memory formation [36], spatial learning [37–39], learned helplessness [40], the effects of social context on learned behavior [41, 42], and individual differences in learning [34]. In addition to behavioral research, there is also substantial work investigating the neurophysiology of cockroaches, with a focus on the mushroom body, a neural structure important to insect learning [43–45]. Topics include distribution of dopaminergic neurons [46], the role of nitric oxide signaling in long term memory [47], the effect of octopamine on feeding behavior [48], and parallel odor processing pathways [49, 50].

Most research on cockroach behavior, including the preceding citations, has been on *Periplaneta americana* or *Blattella germanica*, two well-known domiciliary pests [16]. As both species are excellent climbers, and *P. americana* is also able to fly, some laboratories may have concerns about housing these potential pests. We believe the orange head cockroach, *Eublaberus posticus*, may be a good alternative for laboratories that are concerned about the escape and infestation potential of pest species. *E. posticus* is a large Central and South American species that, unlike *P. americana* and *Blattella germanica*, would have difficulty surviving outside of a tropical environment. Additionally, *E. posticus* is unable to fly or climb smooth surfaces, making escapes rare and simplifying apparatus design. While there are a few studies on the social behavior of *E. posticus* [51, 52], we are only familiar with a single paper on learning in this species; a series of habituation experiments conducted by our laboratory [53].

We believe that *E. posticus* is an ideal alternative to potential pest species, though not necessarily the only alternative. Closely related species in the genera *Blaberus* and *Eublaberus* may also be good candidates as they are tropical species that are generally unable to fly or climb smooth surfaces, with *Blaberus discoidalis* being another well-known, easy to acquire species. *Blaptica dubia* is another popular and related species that shares these practical traits. The common hissing cockroach, *Gromphadorhina portentosa*, is perhaps the best-known non-pest

species. However, unlike *E. posticous*, they are excellent climbers [15]. More importantly, in pilot research, we have found their appetite to be inconsistent. While *G. portentosa* has fascinating vocal and social behavior, we have found their weak appetite makes traditional food-related conditioning procedures challenging. We selected *E. posticous* over other well-known species with similar characteristics, such as *Blaberus discoidalis* and *Blaptica dubia*, as *E. posticus* is known to have a voracious appetite, even predating on smaller insects [15]. Feeding reliability is an important component of many associative conditioning procedures, and the need for reliable feeding behavior was especially salient for us after pilot work with *G. portentosa*.

In this paper, we investigate associative learning and memory in *E. posticus* in two experiments. In our first experiment, we investigate the ability of *E. posticus* to learn to associate an arbitrary conditioned stimulus odor with either an appetitive food solution or an aversive salt-water solution. In our second experiment, we repeat the conditioning methods of the first experiment, but also test for preferences to conditioned odors 15 to 1,215 minutes after conditioning to explore the extent that associative memories are retained.

## Methods

### Subjects

Subjects were adult orange-head cockroaches (*Eublaberus posticus*), maintained in two large plastic colonies (52 x 36 x 36 cm) with a layer of Repti Bark substrate (Zoo Med Laboratories; San Luis Obispo, CA). Founding members of the colonies were acquired through Roach Crossing (roachcrossing.com; Livonia, MI) and Cape Cod Roaches (capecodroaches.com; Boston, MA). Prior to collection, each colony initially contained approximately 75 adults. Colony temperature was maintained at 23˚C with relative humidity levels of 61%. The laboratory was kept on a 12:12 hour day-night cycle.

Outside of research periods, colonies were given dry dog food (Purina One, Nestlé Purina PetCare; St. Louis, MO) and water ad libitum and were provided with occasional apples. During research, colonies were provided with a restricted diet of dry dog food (1–4 grams each weekday) and water ad libitum. The purpose of the restricted diet was to increase the number of food-motivated individuals. To collect motivated subjects, we placed an inaccessible cup of food inside the colony and collected only individuals that immediately approached the cup. This method is modeled after other techniques to ensure collection of motivated subjects found in cockroach and bee conditioning research [24, 54, 55]. During collection, the number of motivated subjects was recorded, and the amount of food provided to the colony was adjusted accordingly. After collection, subjects were weighed and sexed, then individually placed in separate plastic experiment chambers, and given one hour to acclimate.

### Apparatus

The apparatus (see Fig 1) consisted of two parts; one clear plastic experiment chamber for each subject (7.3 x 9.9 x 3.1 cm), and two larger filming units (46 x 22 x 23 cm) with spaces to hold up to four chambers each. The individual chambers were sized to allow subjects to freely move inside the container [55–58], rather than restrain the subjects [24, 31, 54], as while the restrained methods are known to work well with honey bees, other species may perform poorly when restrained [56]. The experiment chambers each had six holes (1 cm diameter) distributed across the top of the chambers. Holes were large enough to allow stimulus administration but did not permit escape. Each filming unit contained four slots that positioned the experiment chambers above angled mirrors, permitting filming the ventral sides of subjects. Barriers between the slots prevented subject interaction. Lights were attached to each filming apparatus, and the inside was painted reflective silver to facilitate a well-lit filming environment.

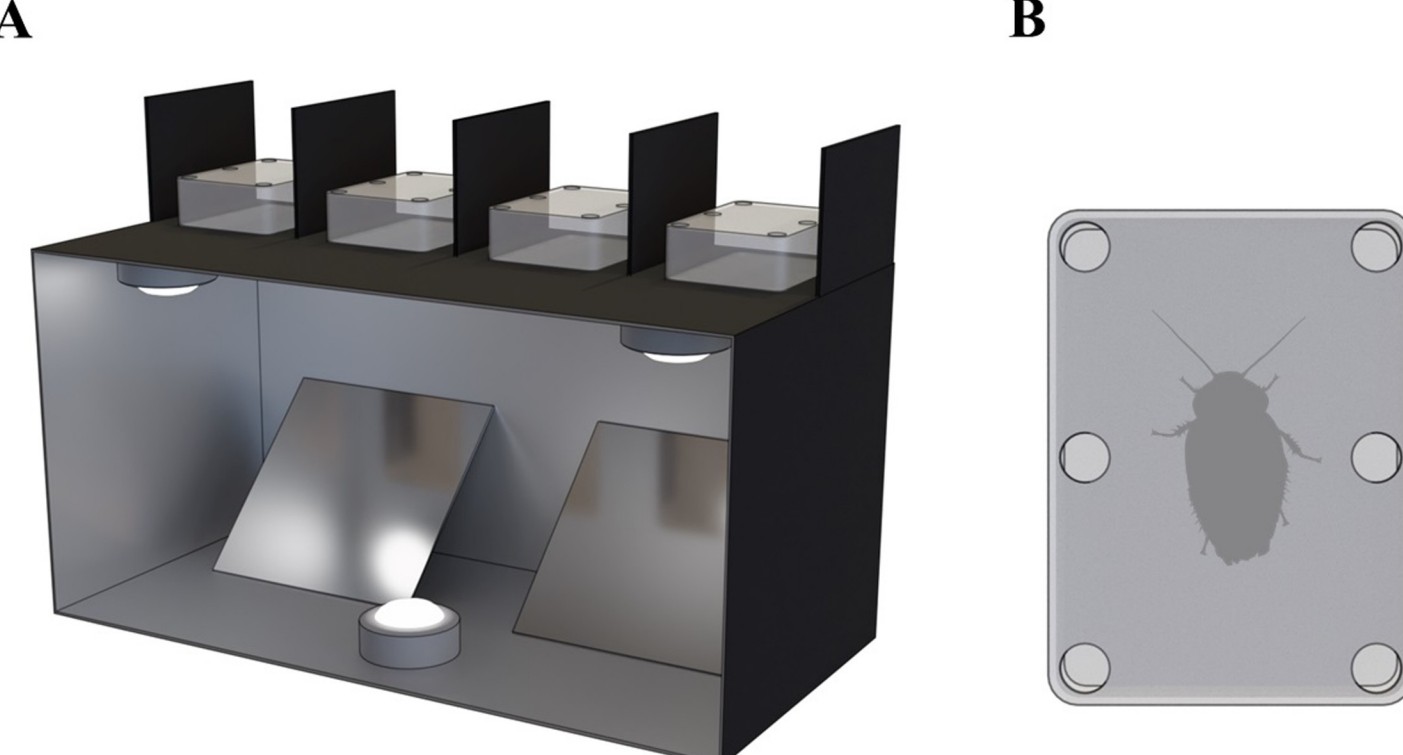

**Fig 1. Apparatus diagram.** (A) Filming unit containing four clear plastic experiment chambers. Three internal lights were used to supplement room lighting. Mirrors were used to film the ventral side of subjects. (B) Top view of one experiment chamber. Six holes in the top of the chambers permitted stimulus administration.

### Stimuli

Several stimuli were presented to subjects via 1 ml syringes through the holes in the tops of the experiment chambers. Unconditioned stimulus (US) solutions were delivered through the syringe, while odors wiped on the surface of the syringe acted as conditioned stimuli (CS). The appetitive unconditioned stimulus (US$^+$) was a strained liquid solution derived from a 20:20:60 (by weight) mixture of dry dog food, sugar, and water. We developed this solution based on the use of sucrose solutions in cockroach and bee research [24, 34, 54, 55, 58], as well as pilot research suggesting the inclusion of dog food was important. Dry ingredients were pulverized in a spice grinder and mixed with hot water until fully incorporated. The contents were strained through cheesecloth to create the final liquid solution. We then colored the solution with a mixture of red and yellow food dye to mask the color of dog food. The aversive unconditioned stimulus (US$^-$) was a 20:80 (by weight) mixture of NaCl and water. We selected this US$^-$ based on other research with cockroaches [55], and because we found this ratio of NaCl and water was aversive but did not inhibit feeding from other solutions in pilot experiments. We colored the mixture with food dye to match the US$^+$. Dying both solutions served to reduce the possibility the subjects could learn visual distinctions between the appetitive and aversive solutions.

Conditioned stimulus odors were orange extract (McCormick Pure Orange Extract; Hunt Valley, MA) and coffee (Publix Brand Instant Coffee; Lakeland, FL; 10:90 coffee:water ratio by weight). The odors were applied to syringes by wiping them with a paper towel fully saturated in the odor solution. We used odors as stimuli due to the prevalence of olfactory conditioning in the insect literature including several experiments with cockroaches [6, 8, 9, 32–36, 54–57].

We selected these odors specifically as our pilot research suggested they were neutral, unlike vanilla and peppermint which may be innately appetitive and aversive, respectively, to some species of cockroach [55].

## Experiment 1

Subjects were 28 male and 36 female cockroaches with average weights of 2.3g and 3.8g, respectively. Subjects were divided into either paired or unpaired groups. The general methods for both groups were based on established honey bee procedures [24, 54], as well as recent cockroach work [32, 33, 36, 55]. Each group experienced a one-hour acclimation followed by 10 conditioning trials, each separated by 15-minute inter-trial intervals (ITI). In the paired group (n = 32), subjects received differential conditioning where each trial either involved an appetitive association of $CS^+$ and $US^+$ or an aversive association of $CS^-$ and $US^-$. For both appetitive and aversive trials, we used a forward delay conditioning procedure with a four-second CS followed by a four-second US with continual CS overlap. In appetitive trials, the $CS^+$ was presented for four seconds by placing a syringe coated in CS odor close to the head of the subject through holes in the top of the experiment chamber. During this time, the $US^+$ solution was present but withdrawn inside the syringe. The $US^+$ was presented after four seconds by suspending a droplet of the $US^+$ solution at the tip of the syringe. The solution was lightly touched to the antennae unless the subject immediately made contact with its mouthparts. This solution presentation method is common to both cockroach and bee conditioning procedures [24, 36, 54]. Aversive trials were similar except that the $CS^-$ and $US^-$ were used in place of the $CS^+$ and $US^+$. Each subject received five appetitive and five aversive trials presented in either an ABAABABBAB or a BABABBABAA pattern, with "A" representing an appetitive trial and "B" representing an aversive trial. The use of orange and coffee as $CS^+$ and $CS^-$ was counterbalanced across subjects.

In the unpaired group (n = 32), subjects received five trials of unpaired appetitive unconditioned stimulus ($US^u$) and five trials of an unpaired conditioned stimulus ($CS^u$), but never received $US^u$ and $CS^u$ in the same trial. This group was used to control for the possibility that a response to a CS could result from sensitization instead of associative learning. Stimuli were presented the same manner as the paired group, except that the $CS^u$ was presented on a syringe containing only colored water instead of a $US^+$ or $US^-$ solution. Half of the subjects received alternating $US^u$ and $CS^u$ trials beginning with the $CS^u$, while the other half began with the $US^u$. The use of orange and coffee as $CS^u$ was counterbalanced across subjects.

Subjects in both groups received a preference test 15 minutes after completion of the 10 conditioning trials. The preference test was conducted by placing two scented syringes filled with colored water in front of the subject. The subject was given one minute to freely interact with or ignore the syringes. This procedure was used in both paired and unpaired groups; the only difference being that, for the unpaired group, the one odor, $CS^n$, was novel.

## Experiment 2

Subjects were 20 male and 30 female cockroaches with average weights of 2.21g and 3.78g, respectively. Subjects were maintained and collected in the same manner as described in Experiment 1. All methods followed those of the paired group of Experiment 1, with two exceptions. First, only the ABAABABBAB pattern was used. Second, subjects were divided into five groups of 10, with each group experiencing a delay of either 15, 45, 135, 405 or 1,215 minutes between the final conditioning trial and the preference test. In this experiment, the preference test acted as an assessment of how the delay interval affected behavior. All subjects

were fed, and were observed eating, immediately after the preference test to ensure that subjects in all groups were similarly active and able to feed after the delay period.

## Analysis

During conditioning trials, we recorded approach scores in response to the US and the CS. The approach scores were derived by noting the subject's position and rotation when a stimulus was presented, and then again when the stimulus ended. We recorded the approach score as either -1: moving or rotating away from the stimulus, 0: no change in position or rotation, or 1: approaching or rotating toward the stimulus. We also recorded maxillary palp extension response (MPR) during both the conditioning trials and the preference test as an analog to the proboscis extension response (PER) commonly used to study bee learning [24]. We recorded both total duration and latency of MPR in response to the US and the CS for conditioning trials, as well as in response to the CS during the preference test. All behaviors were independently scored from video by two observers. A third observer later independently verified a random 20% of trials.

All analyses were conducted through the StatsModels package [59] included in the Anaconda distribution of Python, a free scientific analysis distribution of the Python programming language. We analyzed approach scores and MPR duration for the conditioning trials using a repeated measures regression via generalized estimating equations (GEE) with a Gaussian link [60]. For analyzing the probability of MPR, we used a logistic link. The repeated measures aspect of the GEE regression controlled for repeated or dependent measures from each subject using an exchangeable dependence structure. We used interceptless models where the categorical trial type variable (appetitive, aversive, or unpaired) was treated as three mutually exclusive variables. Individual parameter estimates were then compared directly by creating a z score by dividing the difference between the estimates by the square root of the sum of the squared standard errors of the estimates [61, 62]. The difference between estimates, z score, and *p* value are the same as those normally reported by a regression that includes one level of a categorical variable in the intercept.

For pairwise comparisons, a multiple comparison correction of *p*-values may be useful. The reader is free to select their own significance threshold and adjustment criteria. One simple approach is adjusting the traditional alpha value of 0.05 using a Bonferroni correction, by dividing the alpha value by the number of pairwise comparisons being made. For example, in final three rows of Table 1, we compare the rate that approach scores change across trial. The provided *p*-values could either be compared to the traditional alpha value of 0.05, or a Bonferroni corrected threshold of 0.0167 (0.05 divided by three comparisons). Several other multiple comparison methods could also be used. For the sake of simplicity, we will emphasize the strongest and most significant effects, and note when any important findings could be interpreted differently based on the type of correction used.

## Results

### Experiment 1–Associative learning

In Experiment 1, we investigated the ability of orange head cockroaches to learn appetitive and aversive associations in a differential conditioning procedure, with an unpaired group acting as a control. Then, we tested responses to stimuli in the absence of appetitive or aversive solutions in a preference test.

Fig 2 shows average approach scores for the paired and unpaired groups during the conditioning procedure. Tables 1 and 2 show regression results that correspond to the graphed approach scores. The parameter estimates in the top portions of the tables show the intercepts

**Table 1. Experiment 1 conditioning US approach scores.**

| Parameter | Estimate | Standard Error | 95% Confidence Intervals | | p-value |
|---|---|---|---|---|---|
| Appetitive | 0.475 | 0.141 | 0.199 | 0.751 | 0.001 |
| Aversive | -0.494 | 0.117 | -0.723 | -0.264 | 0.000 |
| Unpaired | 0.631 | 0.107 | 0.421 | 0.842 | 0.000 |
| Appetitive * Trial | 0.037 | 0.041 | -0.043 | 0.118 | 0.362 |
| Aversive * Trial | -0.037 | 0.028 | -0.092 | 0.017 | 0.178 |
| Unpaired * Trial | 0.081 | 0.023 | 0.036 | 0.127 | 0.000 |
| **Pairwise Comparison** | | | **Difference** | **z-score** | **p-value** |
| Appetitive vs. Aversive | | | 0.969 | 5.295 | 0.000 |
| Appetitive vs. Unpaired | | | -0.156 | -0.883 | 0.377 |
| Aversive vs. Unpaired | | | -1.125 | -7.081 | 0.000 |
| Appetitive * Trial vs. Aversive * Trial | | | 0.075 | 1.509 | 0.131 |
| Appetitive * Trial vs. Unpaired * Trial | | | -0.044 | -0.926 | 0.354 |
| Aversive * Trial vs. Unpaired * Trial | | | -0.119 | -3.278 | 0.001 |

and trial interaction slopes for each trial type, while the pairwise comparisons in the bottom portions of the tables show the differences between these intercepts and slopes. Although the figure and tables can be interpreted independently, they are best interpreted together.

Overall, the US approach scores were high for both the appetitive and unpaired trials while they were low for the aversive trials. Most of this trend can be accounted for by the initial US approach scores, with the appetitive and unpaired trials having scores significantly greater than 0 (Table 1; $p$-values < 0.002), and the aversive trial having a score significantly lower than 0 (Table 1; $p$ = 0.000). While a slight increase across trials was observed for the appetitive trials, and a slight decrease was observed for the aversive trials, these effects were not significant. Only the unpaired trials showed a significant increase (Table 1; $p$ = 0.000). This effect is small, however, and is not statistically distinct from the appetitive trial effect (Table 1; $p$ = 0.354). Taken together, the US approach scores confirm the attractive and repellent natures of the solutions we used.

The initial CS approach scores were similar and statistically indistinct across trial types (Table 2; $p$-values > 0.222). However, the appetitive CS approach scores showed a significant increase across trials (Table 2; $p$ = 0.002), while both the aversive and unpaired scores showed significant decreases (Table 2; $p$-values < 0.018), leading to rapid divergence between the appetitive and other trial types. The strong increase in CS approach scores for the appetitive group suggests that the appetitive association was effective at developing a learned response. Additionally, this learning cannot be explained by sensitization alone as the increase was observed in the appetitive, but not unpaired, trials. Interestingly both the intercepts and slopes of the aversive and unpaired trial types were statistically indistinct (Table 2; $p$-values > 0.222).

Fig 3 shows average MPR duration during the conditioning procedure, while Tables 3 and 4 show corresponding regression results. For appetitive trials, the US MPR duration was initially high, with a slight, significant increase over time (Table 3; $p$ = 0.004). Conversely, for the aversive trials, US MPR duration was initially low, with a slight, significant decrease over time (Table 3; $p$ = 0.003). This again illustrates the attractive and repellent natures of these solutions. While US MPR duration for appetitive trials was greater than those for unpaired trials, responses during these trial types were statistically indistinct, both in initial response duration and in change across trial (Table 3; $p$-values > 0.071). The initial CS MPR durations were statistically indistinct across trial types (Table 4; $p$-values > 0.083). However, the appetitive CS$^+$ MPR showed a significant increase (Table 4; $p$ = 0.000), while the aversive CS$^-$ MPR showed a

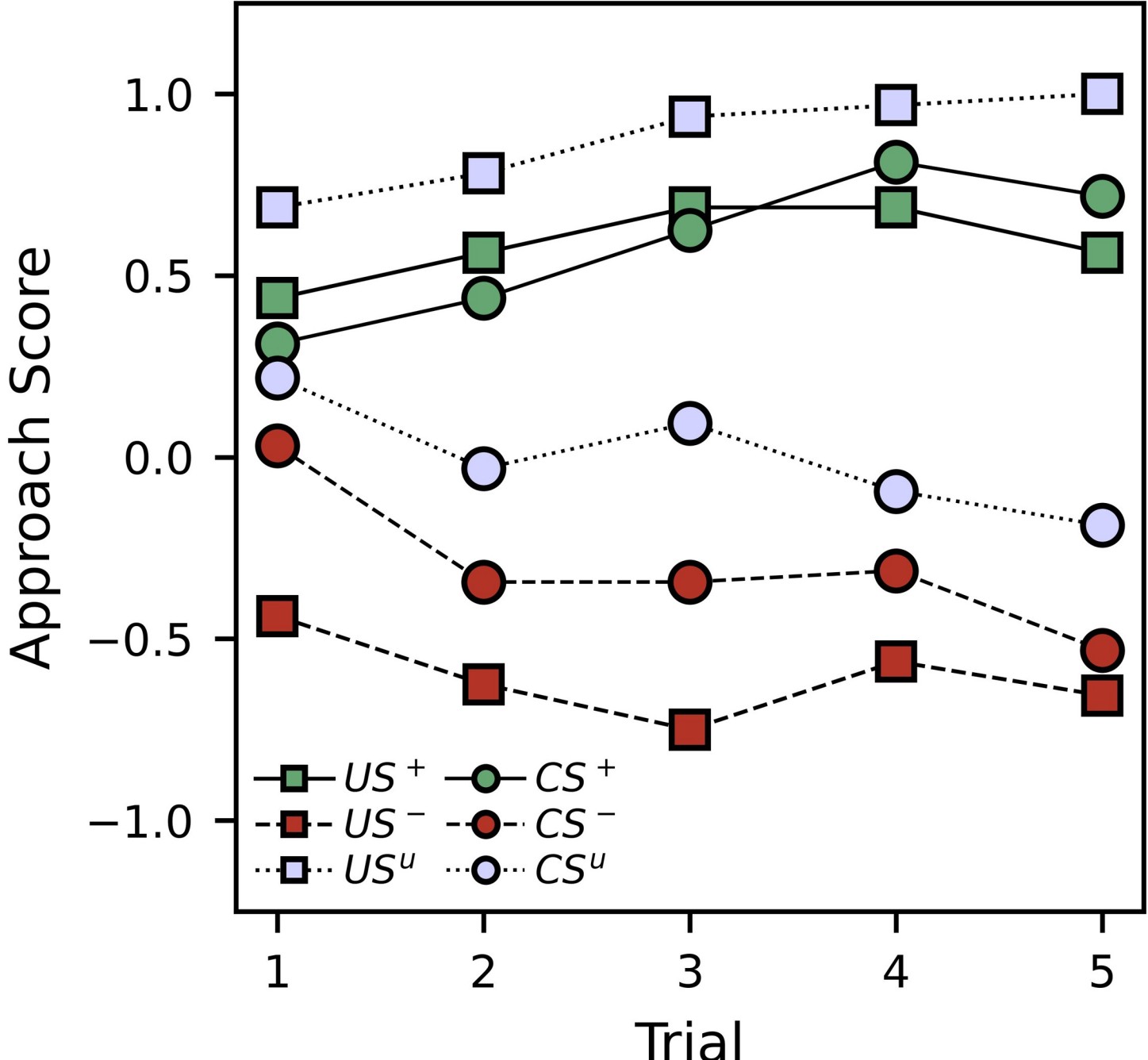

**Fig 2. Experiment 1 average approach scores in response to the US and CS for the paired and unpaired groups, divided trial type, during the conditioning trials.**
Positive approach scores indicate moving or rotating toward a stimulus, negative approach scores indicate moving or rotating away from a stimulus, and zero approach scores indicate no change.

significant decrease (Table 4; $p = 0.001$). Again, the rapid increase in response exclusively to the $CS^+$ suggests that the subjects were learning due to the appetitive association. Interestingly, although the unpaired trials did not show a significant decrease in CS MPR (Table 4; $p = 0.139$), the observed decreasing trend was also not significantly different from that of the aversive trials (Table 4; $p = 0.222$).

**Table 2. Experiment 1 conditioning CS approach scores.**

| Parameter | Estimate | Standard Error | 95% Confidence Intervals | | p-value |
|---|---|---|---|---|---|
| Appetitive | 0.225 | 0.147 | -0.063 | 0.513 | 0.126 |
| Aversive | 0.028 | 0.156 | -0.277 | 0.333 | 0.857 |
| Unpaired | 0.263 | 0.113 | 0.042 | 0.483 | 0.020 |
| Appetitive * Trial | 0.119 | 0.039 | 0.043 | 0.195 | 0.002 |
| Aversive * Trial | -0.109 | 0.035 | -0.178 | -0.040 | 0.002 |
| Unpaired * Trial | -0.088 | 0.037 | -0.159 | -0.016 | 0.017 |
| **Pairwise Comparison** | | | **Difference** | **z-score** | **p-value** |
| Appetitive vs. Aversive | | | 0.197 | 0.919 | 0.358 |
| Appetitive vs. Unpaired | | | -0.037 | -0.202 | 0.840 |
| Aversive vs. Unpaired | | | -0.234 | -1.220 | 0.223 |
| Appetitive * Trial vs. Aversive * Trial | | | 0.228 | 4.350 | 0.000 |
| Appetitive * Trial vs. Unpaired * Trial | | | 0.206 | 3.860 | 0.000 |
| Aversive * Trial vs. Unpaired * Trial | | | -0.022 | -0.431 | 0.667 |

Generally, when subjects began emitting MPR, they continued to emit MPR for the entire stimulus duration. MPR latency was thus highly negatively correlated with MPR duration. Replacing null values of non-responders with the maximum latency value (4 seconds) resulted in regression coefficient values of -0.96 for the US and -0.97 for the CS. Similar trends were observed when dropping the missing latency values or when replacing the values with the mean of latency. Regardless of technique, $p$-values were extremely low ($p < 0.00000$). As MPR latency was effectively the inverse of MPR duration for the conditioning procedure, it will not be discussed further.

Fig 4 shows average MPR duration and latency for the paired and unpaired groups during the preference test. Tables 5 and 6 show the corresponding regression results. For these tables, the pairwise comparisons will be more useful, but we included the full regression for the sake of transparency. For the paired group, MPR duration in response to the appetitive $CS^+$ was significantly greater than that of the aversive $CS^-$ (Table 5; $p = 0.000$). Additionally, the latency to respond to the $CS^+$ was significantly lower (Table 6; $p = 0.000$). The findings for the paired group indicate that subjects truly learned the associations between CS and US and were not simply responding to some faint residual odors of the US when the CS was delivered. For the unpaired group, responses to the novel $CS^n$ were significantly greater in duration than those to the unpaired $CS^u$ (Table 5; $p = 0.000$) and were significantly lower in latency (Table 6; $p = 0.002$). This could be caused by a novelty effect, but may also be a result of habituation or extinction of MPR to the unpaired CS.

Interestingly, the difference in responses between the $CS^+$ and $CS^-$ for the paired group mirror the difference between the $CS^u$ and $CS^n$ for the unpaired group. Indeed, the statistical analysis suggests that response to the $CS^-$ is similar to the $CS^u$ with respect to both duration (Table 5; $p = 0.475$) and latency (Table 6; $p = 0.506$). This similarity is consistent with the CS approach scores (Fig 2), and CS MPR durations (Fig 3) during the conditioning procedure. The similarity between responses to the $CS^+$ and the $CS^n$ are not as clear. Table 5 shows that MPR duration is greater to the paired $CS^+$, but with a $p$-value of 0.026, the significance of this effect is unclear. While a traditional significance threshold of 0.05 considers this finding significant, the pairwise comparisons are ideal candidates for multiple comparisons corrections. A conservative Bonferroni correction leads to a threshold of 0.008 (0.05/6) and a nonsignificant interpretation. With respect to MPR latency, the response to the paired $CS^+$ does appear to be significantly lower than that of the unpaired $CS^n$ (Table 6; $p = 0.003$).

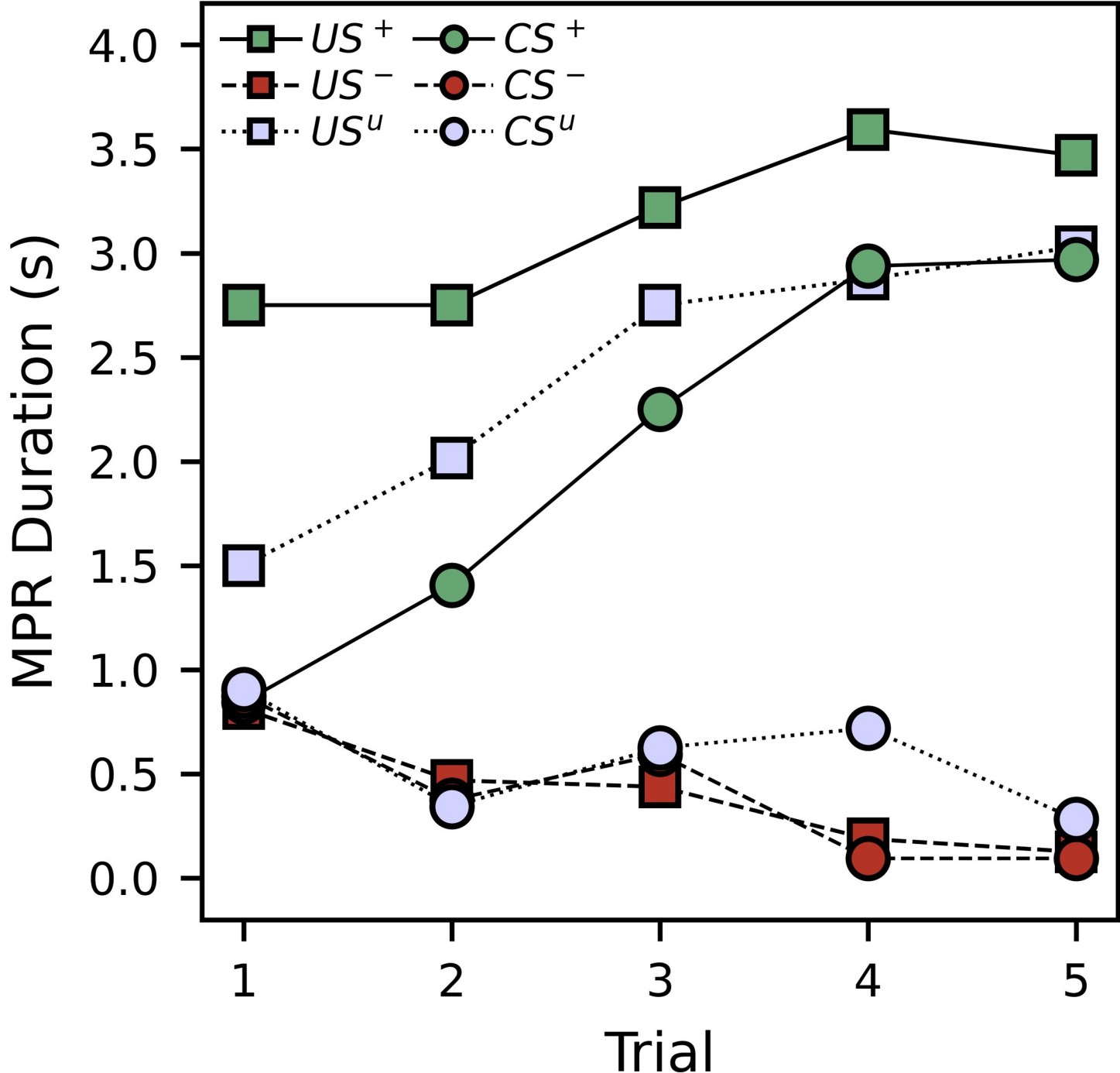

**Fig 3. Experiment 1 average MPR duration in response to the US and CS for the paired and unpaired groups, divided trial type, during the conditioning trials.**

### Experiment 2–Memory

In Experiment 2, we investigated the effect of time delays on performance in the post-conditioning preference test. While we were primarily interested in the preference test, we included figures and analysis for the conditioning sessions in the S1 File. Performance during conditioning was very similar to what we observed during Experiment 1 and will not be discussed further.

**Table 3. Experiment 1 conditioning US MPR duration.**

| Parameter | Estimate | Standard Error | 95% Confidence Intervals | | p-value |
|---|---|---|---|---|---|
| Appetitive | 2.472 | 0.336 | 1.813 | 3.131 | 0.000 |
| Aversive | 0.903 | 0.256 | 0.400 | 1.406 | 0.000 |
| Unpaired | 1.258 | 0.204 | 0.858 | 1.657 | 0.000 |
| Appetitive * Trial | 0.228 | 0.080 | 0.071 | 0.385 | 0.004 |
| Aversive * Trial | -0.166 | 0.055 | -0.274 | -0.058 | 0.003 |
| Unpaired * Trial | 0.392 | 0.044 | 0.306 | 0.478 | 0.000 |
| **Pairwise Comparison** | | | **Difference** | **z-score** | **p-value** |
| Appetitive vs. Aversive | | | 1.569 | 3.711 | 0.000 |
| Appetitive vs. Unpaired | | | 1.214 | 3.089 | 0.002 |
| Aversive vs. Unpaired | | | -0.355 | -1.083 | 0.279 |
| Appetitive * Trial vs. Aversive * Trial | | | 0.394 | 4.055 | 0.000 |
| Appetitive * Trial vs. Unpaired * Trial | | | -0.164 | -1.800 | 0.072 |
| Aversive * Trial vs. Unpaired * Trial | | | -0.558 | -7.925 | 0.000 |

Fig 5 shows average MPR duration for all groups during the preference test. Corresponding regression results can be seen in Table 7. MPR duration in response to the appetitive $CS^+$ was low 15 minutes after the conditioning procedure ended, peaked 45 minutes after conditioning, then decreased with time. Though the response duration at 15 minutes was low, the analysis suggests an overall linear decrease in response duration as a function of time ($p = 0.000$). Out of 50 subjects, 33 responded to the $CS^+$, but only 5 responded to the $CS^-$. Responses to the $CS^-$ were consistently low in duration and were initially significantly lower than responses to the $CS^+$ ($p = 0.000$).

Fig 6 shows the average MPR latency during the preference test. Generally, responses latencies to CS+ were much lower, with an average of 9.88 seconds, than response latencies to CS-, with an average of 41.00 seconds. Given that few subjects responded to the CS-, we do not believe further discussion and analysis of latency will be useful. Curious readers may view our initial analysis of latency in the S2 File, though given the nature of the data it should be interpreted with caution.

**Table 4. Experiment 1 conditioning CS MPR duration.**

| Parameter | Estimate | Standard Error | 95% Confidence Intervals | | p-value |
|---|---|---|---|---|---|
| Appetitive | 0.347 | 0.271 | -0.185 | 0.879 | 0.201 |
| Aversive | 0.959 | 0.228 | 0.513 | 1.406 | 0.000 |
| Unpaired | 0.838 | 0.237 | 0.373 | 1.302 | 0.000 |
| Appetitive * Trial | 0.578 | 0.078 | 0.426 | 0.730 | 0.000 |
| Aversive * Trial | -0.184 | 0.053 | -0.288 | -0.080 | 0.001 |
| Unpaired * Trial | -0.087 | 0.059 | -0.203 | 0.028 | 0.139 |
| **Pairwise Comparison** | | | **Difference** | **z-score** | **p-value** |
| Appetitive vs. Aversive | | | -0.612 | -1.729 | 0.084 |
| Appetitive vs. Unpaired | | | -0.491 | -1.362 | 0.173 |
| Aversive vs. Unpaired | | | 0.122 | 0.371 | 0.711 |
| Appetitive * Trial vs. Aversive * Trial | | | 0.762 | 8.108 | 0.000 |
| Appetitive * Trial vs. Unpaired * Trial | | | 0.666 | 6.822 | 0.000 |
| Aversive * Trial vs. Unpaired * Trial | | | -0.097 | -1.220 | 0.222 |

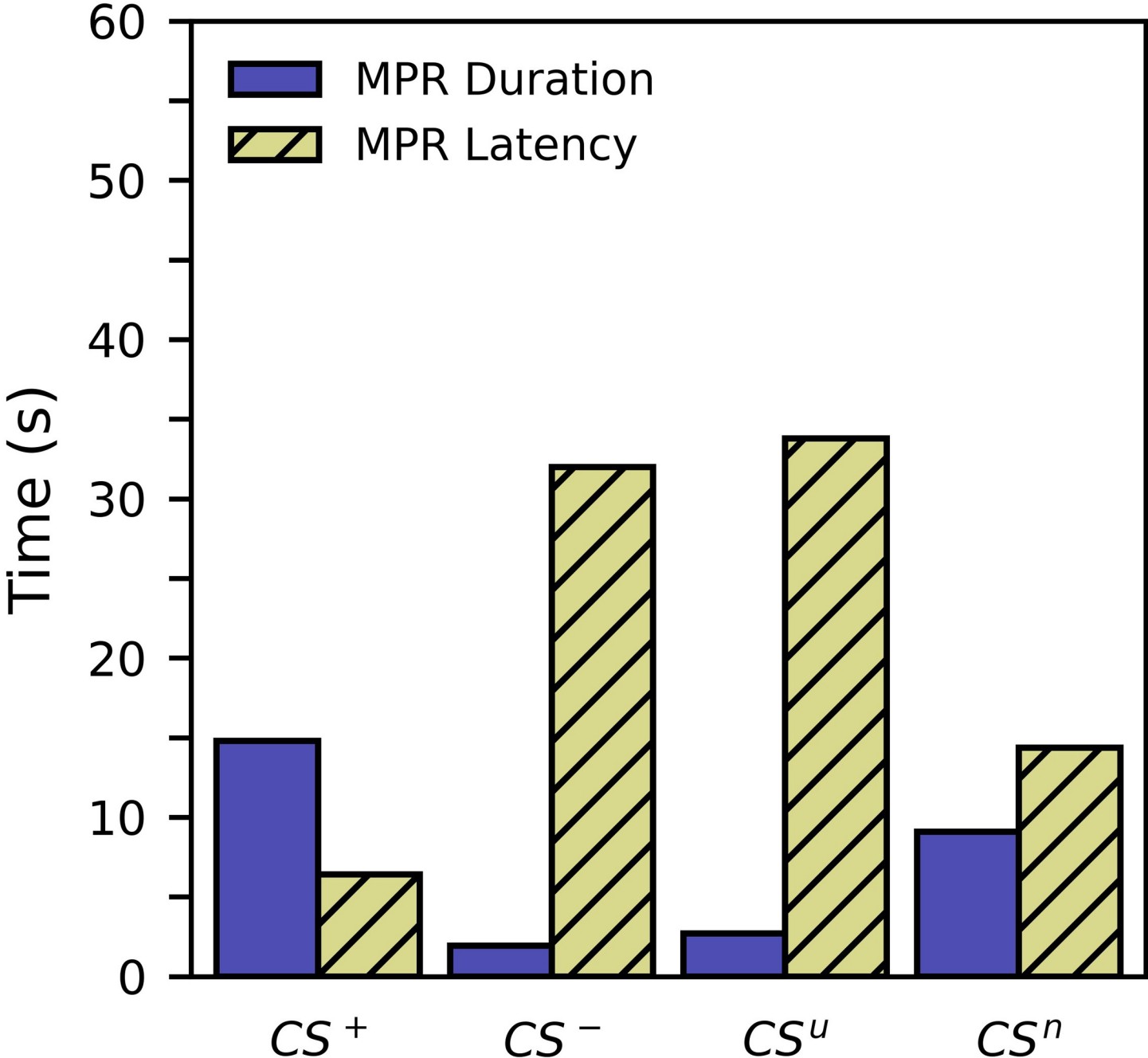

**Fig 4. Experiment 1 average MPR duration and latency for the paired and unpaired groups during the preference test.** The number of subjects responding to the $CS^+$, $CS^-$, $CS^u$ and $CS^n$ were 25, 10, 13, and 20, respectively. Averaged latency values exclude non-responders.

Table 8 shows analysis for the probability of responding to the $CS^+$ or $CS^-$. For the appetitive $CS^+$, the response probability was initially significantly greater than chance (50%; $p = 0.000$), and decreased slightly, but significantly, as a function of time ($p = 0.001$). Responding to the aversive CS- was significantly less than chance ($p = 0.018$), and also decreased as a function of time, but this effect was only borderline significant ($p = 0.049$). Pairwise comparisons suggest that the decrease with time was statistically similar for both stimuli ($p = 0.527$).

**Table 5. Experiment 1 preference MPR duration.**

| Parameter | Estimate | Standard Error | 95% Confidence Intervals | | *p*-value |
|---|---|---|---|---|---|
| Paired appetitive | 14.781 | 1.959 | 10.942 | 18.621 | 0.000 |
| Paired aversive | 1.938 | 0.737 | 0.493 | 3.382 | 0.009 |
| Unpaired | 2.688 | 0.747 | 1.224 | 4.151 | 0.000 |
| Unpaired novel | 9.094 | 1.635 | 5.888 | 12.299 | 0.000 |
| **Pairwise Comparison** | | | **Difference** | ***z*-score** | ***p*-value** |
| Paired appetitive vs. Paired aversive | | | 12.844 | 6.137 | 0.000 |
| Paired appetitive vs. Unpaired | | | 12.094 | 5.769 | 0.000 |
| Paired appetitive vs. Unpaired novel | | | 5.688 | 2.229 | 0.026 |
| Paired aversive vs. Unpaired | | | -0.750 | -0.715 | 0.475 |
| Paired aversive vs. Unpaired novel | | | -7.156 | -3.990 | 0.000 |
| Unpaired vs. Unpaired novel | | | -6.406 | -3.563 | 0.000 |

## Discussion

Our work is the first to show associative learning in the orange head cockroach (*Eublaberus posticus*), and the second to show any form of learning in this species [53]. While there are several experiments demonstrating associative learning in cockroaches, few use methods and analysis that permit specific comparisons of acquisition trends. For example, many excellent experiments using *Periplaneta americana* report learning as a change in distribution of preference between odors, but do not report average acquisition curves easily comparable to our work [55]. One exception, however, does show acquisition curves in *P. americana* that are very similar to what we observed in *E. posticus* [36]. Interestingly, our acquisition curves are also similar to those observed in the honey bee proboscis extension response conditioning work that was a major influence for our methods [10, 24, 63].

While associative learning is studied less in cockroaches than in other popular insect models, we are aware of demonstrations in four species: *P. americana* [36, 55], *Blattella germanica* [57], *Blatta orientalis* [64], and *Rhyparobia madare* [35, 65]. Habituation, but not associative conditioning, has also been observed in *Gromphadorina portentosa* [66]. Most of this work has been conducted with *P. americana*, a member of the superfamily Blattoidae, which also

**Table 6. Experiment 1 preference MPR latency.**

| Parameter | Estimate | Standard Error | 95% Confidence Intervals | | *p*-value |
|---|---|---|---|---|---|
| Paired appetitive | 8.880 | 1.372 | 6.191 | 11.568 | 0.000 |
| Paired aversive | 22.193 | 2.252 | 17.779 | 26.607 | 0.000 |
| Unpaired | 24.249 | 2.119 | 20.096 | 28.402 | 0.000 |
| Unpaired novel | 15.619 | 1.805 | 12.083 | 19.156 | 0.000 |
| **Pairwise Comparison** | | | **Difference** | ***z*-score** | ***p*-value** |
| Paired appetitive vs. Paired aversive | | | -13.313 | -5.049 | 0.000 |
| Paired appetitive vs. Unpaired | | | -15.369 | -6.089 | 0.000 |
| Paired appetitive vs. Unpaired novel | | | -6.740 | -2.973 | 0.003 |
| Paired aversive vs. Unpaired | | | -2.056 | -0.665 | 0.506 |
| Paired aversive vs. Unpaired novel | | | 6.574 | 2.278 | 0.023 |
| Unpaired appetitive vs. Unpaired novel | | | 8.630 | 3.101 | 0.002 |

*Note*. Null values of latency were replaced with the mean of latency.

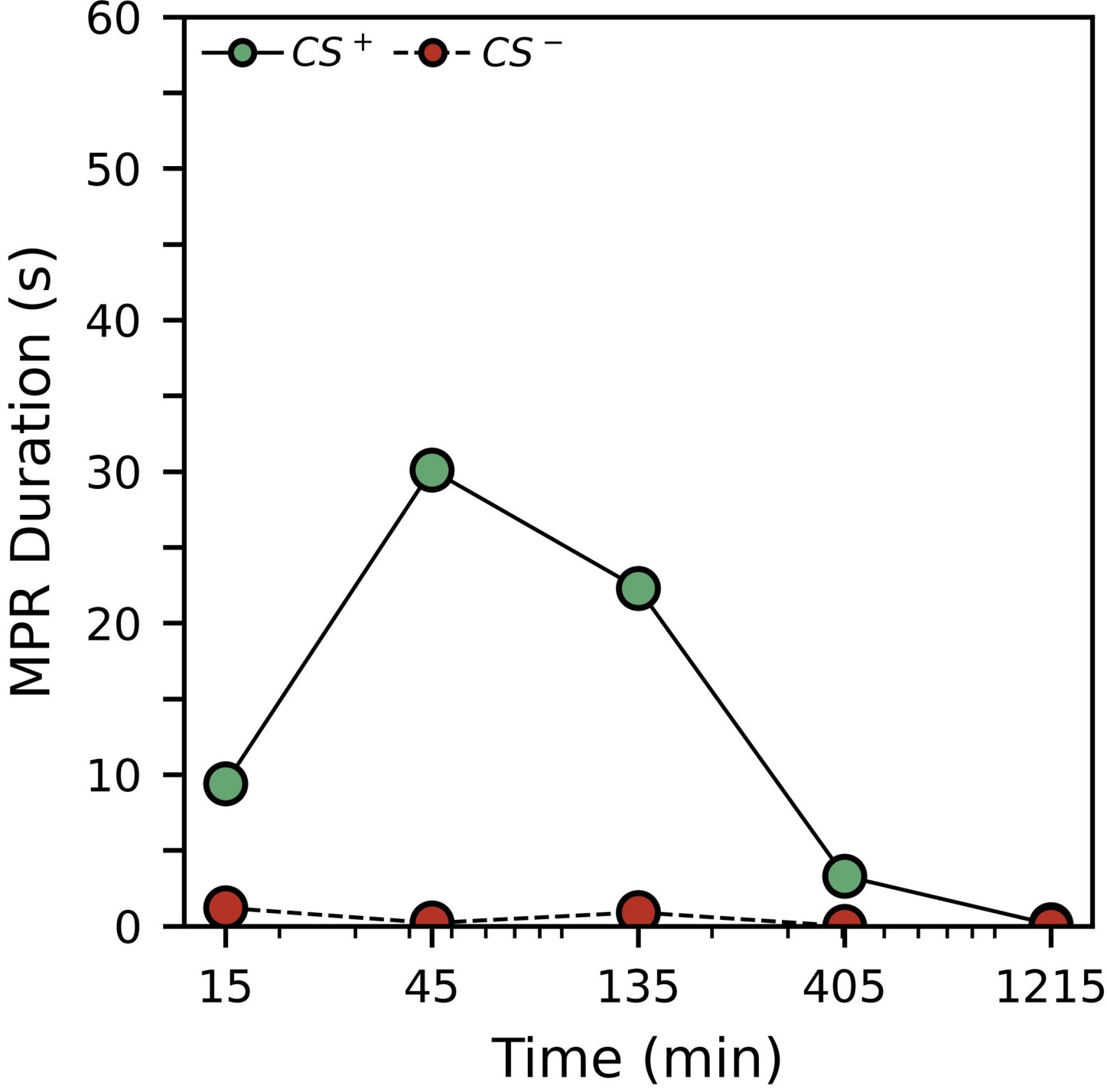

**Fig 5. Experiment 2 average MPR duration in response to the CS$^+$ and CS$^-$ during the preference test.**

includes *Blatta orientalis*. Our *E. posticus*, as well as the other species mentioned, are members of the superfamily Blaberoidae, and more closely related to each other than the popular subject, *P. americana*. Though there is more work to be done to establish a consensus, it is possible that the presence of similar conditioning findings across unrelated cockroach taxa indicate

**Table 7. Experiment 2 preference MPR duration.**

| Parameter | Estimate | Standard Error | 95% Confidence Intervals | | p-value |
|---|---|---|---|---|---|
| Appetitive | 19.454 | 2.669 | 14.223 | 24.686 | 0.000 |
| Aversive | 0.709 | 0.330 | 0.061 | 1.356 | 0.032 |
| Appetitive * Time | -0.018 | 0.003 | -0.023 | -0.013 | 0.000 |
| Aversive * Time | -0.001 | 0.000 | -0.001 | 0.000 | 0.038 |
| **Pairwise Comparison** | | | Difference | z-score | p-value |
| Appetitive vs. Aversive | | | 18.745 | 6.970 | 0.000 |
| Appetitive * Time vs Aversive * Time | | | -0.017 | -5.667 | 0.000 |

that associative learning is a conserved trait common to insects of the order Blattodea, which includes all cockroaches and termites [67].

While our work clearly shows that *E. posticus* has strong associative learning ability consistent with other cockroaches and with honey bees, our findings regarding memory stand out compared to other cockroach work. In our experiment, we saw a nonlinear retention trend, where preference for the conditioned stimulus was moderate 15 minutes after training, increased 45 minutes after training, then decreased as a function of time. Research from other laboratories often produces greater retention. Most notably, *P. americana* has shown preferences being retained even four weeks after training [55]. Retention for at least 24 hours is not uncommon to *P. americana* work. One multi-experiment paper [36] found that around 50% of *P. americana* retained associations after 24 hours in several experiments with varying methods. However, one of the experiments found a similar nonlinear trend to our work; retention was lower 10 and 15 minutes after training, peaked 30 minutes after training, then decreased again. An experiment with *Blattella germanica* found retention 30 minutes after training, less pronounced retention 1 day after training, and no retention 2 days after training [57]. In *R. maderae*, retention was found 2 days to 9 days later, depending on the type of conditioning procedure used [35, 65].

While species is the most immediate difference between our research and that of other work showing greater retention, we do not believe this is the critical factor. A careful consideration of the methods of other experiments reveals two factors that may be more important than species. First, while our subjects received a small amount of their standard food each weekday and received water ad libitum, experiments from other laboratories used partial or full deprivation procedures lasting between two and seven days [35, 36, 55, 57]. Second, while our research was conducted in a well-lit apparatus under standard room lighting during the light phase of a day/night cycle, other research was conducted during the dark phase [36, 55], or during the light phase, but under a dim red light which is not visible to cockroaches [57]. The effect of circadian rhythm on learning and memory has been tested specifically in *R. maderae*; circadian rhythm may affect learning, but not recall in some procedures, while in others recall may be affected, but learning is not [35, 65]. Subsequent memory work with *E. posticus* and other species should test the effects of deprivation and circadian patterns specifically.

A final area of consideration is potential adaptation of our methods for the classroom. Several laboratories advocate use of insects to demonstrate psychological principles due to their inexpensive, practical nature [68, 69], and *E. posticus* certainly fits these criteria. Our method is especially suited to the classroom as *E. posticus* cannot fly or climb and can be worked with in open containers, unlike *P. americana*. Our data also suggests *E. posticus* learns quickly even under the well-lit conditions that would be needed for student observation. Unlike other cockroach and bee preparations, *E. posticus* does not require substantial

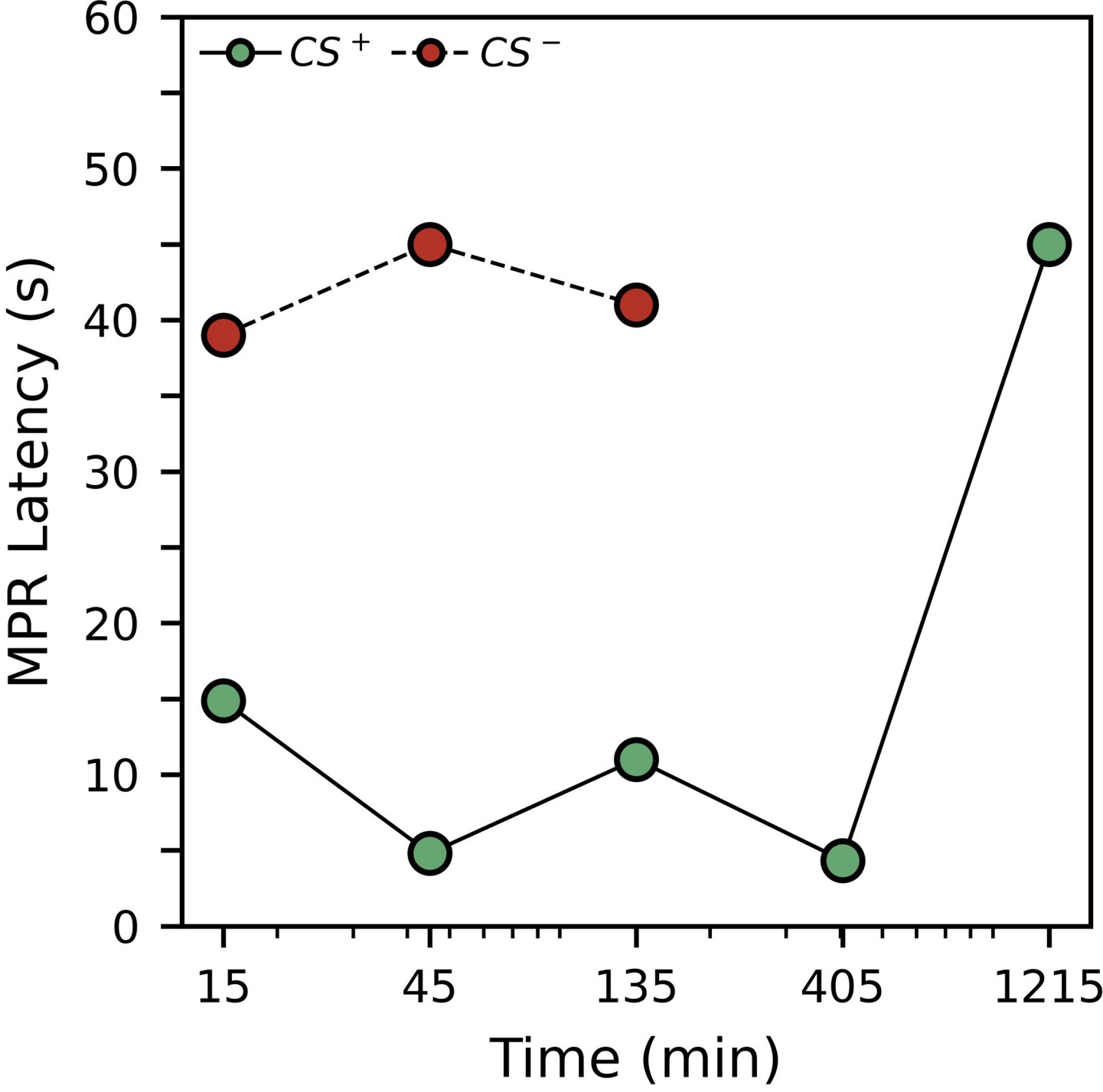

**Fig 6. Experiment 2 average MPR latency to respond to the CS⁺ and CS⁻ during the preference test.** Averaged values exclude non-responders. For the $CS^+$, the sample sizes were 7, 9, 8, 6 and 1 for 15-, 24-, 135-, 405- and 1,215-minute delay groups, respectively. For the $CS^-$, the sample sizes were 2, 1, 2, 0.

deprivation, allowing it to be easily integrated into classroom activities as opportunities permit. We hope that future work considers use of *E. posticus* conditioning procedures both for research and for teaching.

**Table 8. Experiment 2 preference MPR probability.**

| Parameter | Estimate | Standard Error | 95% Confidence Intervals | | p-value |
|---|---|---|---|---|---|
| Appetitive | 2.037 | 0.505 | 1.048 | 3.027 | 0.000 |
| Aversive | -1.318 | 0.557 | -2.409 | -0.227 | 0.018 |
| Appetitive * Time | -0.004 | 0.001 | -0.006 | -0.001 | 0.001 |
| Aversive * Time | -0.006 | 0.003 | -0.012 | 0.000 | 0.049 |
| **Pairwise Comparison** | | | Difference | z-score | p-value |
| Appetitive vs. Aversive | | | 3.355 | 4.462 | 0.000 |
| Appetitive * Time vs Aversive * Time | | | 0.002 | 0.632 | 0.527 |

## Supporting information

**S1 File. Experiment 2 supplementary results.**
(PDF)

**S2 File. Data.**
(ZIP)

## Author Contributions

**Conceptualization:** Christopher A. Varnon, Erandy I. Barrera, Isobel N. Wilkes.

**Formal analysis:** Christopher A. Varnon.

**Funding acquisition:** Christopher A. Varnon.

**Investigation:** Christopher A. Varnon, Erandy I. Barrera, Isobel N. Wilkes.

**Methodology:** Christopher A. Varnon, Erandy I. Barrera, Isobel N. Wilkes.

**Project administration:** Christopher A. Varnon.

**Supervision:** Christopher A. Varnon.

**Writing – original draft:** Christopher A. Varnon.

**Writing – review & editing:** Christopher A. Varnon.

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
