## [Decision Letter · Decision Letter 0]

13 May 2022

PONE-D-22-06736Learning and Memory in the Orange Head Cockroach (Eublaberus posticus)PLOS ONE

Dear Dr. Varnon,

Thank you for submitting your manuscript to PLOS ONE. After careful consideration, we feel that it has merit but does not fully meet PLOS ONE’s publication criteria as it currently stands. Therefore, we invite you to submit a revised version of the manuscript that addresses the points raised during the review process.

We have now received two reviews of your submitted paper, and both reviewers support the publication of your study and suggest several corrections and changes. Specifically, there are no concerns with your actual experiments, except for the unequal sample size, which you should address.

Reviewer #1 has concerns with your rational advocating for *Eublaberus posticus* cockroaches. While you point out that these cockroaches are less likely to escape compared to *Periplaneta americana *and *Blattella germanica*, there are other species such as in the genera *Blaberus *or* Gromphadorhina*, which are very well studied, and which you should discuss and include in your argument. While promoting *Eublaberus posticus *is fine, you should discuss and compare the overall advantages and disadvantages of different cockroach species for research and education.

You should take into account all the reviewers' suggestions and include a figure of your experimental setup, which is difficult to envision from your  methods description.  

We look forward to receiving your revised manuscript.

Kind regards,

Wulfila Gronenberg

Academic Editor

PLOS ONE

Journal Requirements:

Reviewers' comments:

Reviewer's Responses to Questions

**Comments to the Author**

1. Is the manuscript technically sound, and do the data support the conclusions?

Reviewer #1: Yes

Reviewer #2: Yes

2. Has the statistical analysis been performed appropriately and rigorously? 

Reviewer #1: Yes

Reviewer #2: Yes

3. Have the authors made all data underlying the findings in their manuscript fully available?

Reviewer #1: Yes

Reviewer #2: Yes

4. Is the manuscript presented in an intelligible fashion and written in standard English?

Reviewer #1: Yes

Reviewer #2: Yes

5. Review Comments to the Author

Reviewer #1: The authors utilized learning paradigms and implemented the behavior with the orange

orange head cockroach ( Eublaberus posticus ), which has not been demonstrated. Further, the authors state that the goal of this project is to use the orange head cockroach as a model organism for behavioral research. The title and stating the goal definitely sets the stage, and offer a reason as to "why" the orange head cockroach is advantageous as a model organism. I was expecting a myriad of early literature (not cited) elucidating the utilization of cockroaches for learning and memory. I would make sure that if the intro sets the stage, it must be followed by advantages / disadvantages. Early on, there is mention about escape. How so? Is the claim that cockroaches are useful model organisms? is the claim that orange head cockroach is a better species over other cockroaches? I am not convinced. I like how the intro sets up the stage, yet it is never explicitly mentioned.

I would build upon the initial idea about the use of invertebrates as models organisms to study specific aspects of disease or health are available. There is lacking how the link to the orange head cockroach is important and how so over other cockroaches? Are there benefits ? (list them) over others that show impressive behavioral abilities.

There is a disconnect, at least in my mind, since it is not spelled out by the authors in my view. It was not clear why the experimental design and methods were chosen (odor, other than mentioning associative learning) and why and how these are suited for the orange head cockroach? This needs to be developed.

The study is the first to show associative learning and memory in the orange head cockroach. Authors compare in a general way to other invertebrate models as well as to other cockroach research.

Reviewer #2: The manuscript nicely describes experiments investigating learning and memory in the orange head cockroach. The authors describe the advantages of this species compared to other cockroach species well and have a good knowledge of the existing prominent work in insect olfactory learning overall.

Experiments and analyses are well described and easy to understand.

I only have the following minor points:

• Line 119: Since the apparatus was not described before in another publication which can be referred to, a figure with a graph of the apparatus and the different chambers would be nice

• Line 141: if I understand correctly, dye was applied to equal the color for the appetitive and aversive solutions?

• Line 282: add for before unpaired trials

• Line 326: I find the sample size quite different. It would be nicer for comparison to have approximately equal sample size

• Line 400: correct proboscis

• Line 422: add of before the experiments

6. PLOS authors have the option to publish the peer review history of their article (what does this mean?). If published, this will include your full peer review and any attached files.

Reviewer #1: No

Reviewer #2: **Yes: **Nina Deisig

---

## [Author Response · Author response to Decision Letter 0]

17 Jun 2022

We appreciate the efforts of the editor and reviewer. We have made several revisions described in the attached letter. We believe we have addressed all the suggestions, and the manuscript is improved as a result.

---

## [Decision Letter · Decision Letter 1]

25 Jul 2022

Learning and Memory in the Orange Head Cockroach (Eublaberus posticus)

PONE-D-22-06736R1

Dear Dr. Varnon,

We’re pleased to inform you that your manuscript has been judged scientifically suitable for publication and will be formally accepted for publication once it meets all outstanding technical requirements.

Kind regards,

Wolfgang Blenau

Academic Editor

PLOS ONE

Additional Editor Comments (optional):

Reviewers' comments:

Reviewer's Responses to Questions

**Comments to the Author**

1. If the authors have adequately addressed your comments raised in a previous round of review and you feel that this manuscript is now acceptable for publication, you may indicate that here to bypass the “Comments to the Author” section, enter your conflict of interest statement in the “Confidential to Editor” section, and submit your "Accept" recommendation.

Reviewer #1: All comments have been addressed

2. Is the manuscript technically sound, and do the data support the conclusions?

Reviewer #1: Yes

3. Has the statistical analysis been performed appropriately and rigorously? 

Reviewer #1: Yes

4. Have the authors made all data underlying the findings in their manuscript fully available?

Reviewer #1: Yes

5. Is the manuscript presented in an intelligible fashion and written in standard English?

Reviewer #1: Yes

6. Review Comments to the Author

Reviewer #1: This manuscript aims to establish the orange head cockroach as a model organism for behavioral research. The classic work of Charles Turner isn't discussed and this is quite disappointing in particular in light of the experimental design used in this study which is associative learning. I would encourage authors to cite Turner as he was truly a pioneer of of cockroach associative learning, individual differences all while he worked at a High School due to the his color of skin. This is not only a teaching moment for the community in terms of diversity but it is directly linked to the introduction and discussion of this study.

The authors claim that the orange head cockroach is optimal as it is unlikely to escape and thus become a pest. This claim isn't the most compelling in particular as as there are other species of cockroaches (ex: Blaptica dubia, Gromphadorhina portentosa) that similarly are not likely to escape. Authors describe how other species can also be used due to their inability to escape and become pests.

Methods and Experimental design are creative and appropriate, including statistical analyses. The edits in these sections are vey useful including the apparatus diagram, stimuli used and appropriate citations.

Results and Discussion:

The authors here show a compelling associative learning model in orange head cockroaches. It is noteworthy to mention that associative learning studies are not common in cockroaches as in other popular insect models.

Although habituation has also been observed in Gromphadorina portentosa, and other cockroach work in learning and memory, there is more work to be done to establish a consensus, it is possible that the presence of similar conditioning findings across unrelated cockroach taxa indicate that associative learning is a conserved trait common to insects of the order Blattodea, which includes all cockroaches and termites.

Authors discuss potential differences for the discrepancies in the literature in terms of memory retention, duration of labile memory (all at the behavioral level), including circadian rhythms. Further, authors recommend the use of the orange head cockroach in the classroom for educational purposes.

7. PLOS authors have the option to publish the peer review history of their article (what does this mean?). If published, this will include your full peer review and any attached files.

Reviewer #1: No

---

## [Editor Report · Acceptance letter]

28 Jul 2022

PONE-D-22-06736R1 

Learning and Memory in the Orange Head Cockroach (*Eublaberus posticus*) 

Dear Dr. Varnon:

I'm pleased to inform you that your manuscript has been deemed suitable for publication in PLOS ONE. Congratulations! Your manuscript is now with our production department. 

Kind regards, 

on behalf of

Dr. Wolfgang Blenau 

Academic Editor

PLOS ONE